# From Worst-Case Scenarios to Extreme Value Statistics: Local Counterfactuals in Flood Frequency Analysis

Paul Voit<sup>1</sup>, Felix Fauer<sup>2</sup>, and Maik Heistermann<sup>1</sup>

<sup>1</sup>Institute for Environmental Sciences and Geography, University of Potsdam, Potsdam, Germany

**Correspondence:** Paul Voit (voit@uni-potsdam.de)

#### Abstract.

Many aspects of flood risk management require flood frequency analysis (FFA) which is, however, often limited by short observational records - especially for flash floods in small basins. In order to address this issue, we propose to extend the underlying data by local counterfactual scenarios. To that end, heavy precipitation events (HPEs) from nearby, hydrologically similar catchments are used to simulate flood peaks which are then included in the FFA for the catchment of interest. In order to demonstrate the added value of this approach, we used 23 years of radar-based precipitation and a hydrological model, fitted the Generalized Extreme Value (GEV) distribution to three different datasets - observed peaks, counterfactual peaks, and their combination -, and evaluated the resulting three GEV fits by means of the quantile skill score (QSS). For a sample of more than 13,000 German headwater catchments, we could show that local counterfactuals improved quantile estimation, with the level of improvement increasing with return period. The improvement declines when the radius of the transposition domain is extended beyond 30 km. Overall, our results provide a tangible perspective to enhance traditional FFA, producing narrower confidence intervals and more robust estimates for design floods and risk assessments.

#### 1 Introduction

15 Flood frequency analysis (FFA) addresses the probability of floods of a given magnitude and their expected recurrence. It provides the statistical basis for defining extreme events and is, since decades, fundamental for various aspects of flood risk management (Klemes, 1993; Merz and Blöschl, 2008), such as the design of hydraulic infrastructure, landscape planning, flood insurance and many more.

Typically, extremes are extracted from observations (of, e.g., discharge or precipitation accumulated over specific durations) via the block maxima or peak-over-threshold method, and a probability distribution is fitted (Gumbel, 1958), most commonly the Generalized Extreme Value (GEV) or Generalized Pareto (GP) distribution. This allows extrapolation of the tail beyond observed records (Cooley, 2012) and an estimation of occurrence probabilities (return periods) for unobserved events.

<sup>&</sup>lt;sup>2</sup>Institute for Meteorology, Freie Universität Berlin, Berlin, Germany

50

55

However, extreme floods occur rarely (by definition), resulting in limited sample sizes for FFA. Short observational records can increase the sampling error and thus the uncertainty of distribution fitting. Consequently, estimates of occurrence probability are often highly uncertain, which can lead to a severe misrepresentation of risk. This problem is amplified in the case of flash floods: these floods are characterized by a rapid onset and carry high sediment and debris loads, which makes them a highly destructive natural hazard (Barredo, 2007; Llasat et al., 2010; Petrucci et al., 2019; CRED/UCLouvain, 2023). The corresponding rainfall events are typically brief and intense and occur at small spatial scales. They only trigger a flash flood in case they coincide with basins that are able to convert rainfall into runoff, and rapidly propagate that runoff towards the outlet. These processes are governed by topography, geomorphology, soils, and land use. Furthermore, flash-flood—prone basins are generally small (

- factual peak. Flood peak observed in the CoI or modelled with observed rainfall.
- counterfactual peak: Flood peak simulated by a hydrological model forced with a transposed (counterfactual) HPE.

The central issue with approaches such as regionalization, PMP, or SST is the plausibility of counterfactuals. Hazard assessments based on such methods are only meaningful if the counterfactuals are considered realistic. To this end, previous studies have proposed different methods to define a TD from which HPEs are sampled. Fontaine and Potter (1989) described it as a region where "significant storms are uniformly distributed in space," while Zhou et al. (2019) suggested using cloud-to-ground lightning analyses. However, the outcome of these methods also depends on the length of the observed data. Instead of defining a TD based on a complex analysis, Voit and Heistermann (2024a) introduced the concept of local counterfactuals, where counterfactual floods were generated by selecting HPEs that caused high runoff peaks in basins from a close neighborhood and forcing a runoff model with these transposed HPEs. Even within this very small TD and based on only 23 years of data, local counterfactuals produced flood peaks comparable to a 200-year "extreme" flood.

But how can counterfactuals be incorporated into FFA? For numerous application contexts, return periods and design levels remain essential for stakeholders. SST provides one way to statistically assess the occurrence of hypothetical flood scenarios, but it requires both the definition of the TD and the selection of the most relevant rainfall duration for sampling events. The latter is not trivial, as the duration of extreme rainfall that generates the highest flood peak may vary between catchments and is often difficult to determine in advance. For this reason Voit and Heistermann (2024a) proposed a bottom-up approach by selecting the HPEs which caused high flood peaks nearby catchments, irrespective of rainfall duration.

In this study, we propose to extend the concept of local counterfactuals in order to formally integrate counterfactual flood peaks into flood frequency analysis (FFA). This is demonstrated in a case study on the basis of 23 years of radar-based precipitation records in Germany, in combination with a Germany-wide flash flood model as introduced by Voit and Heistermann (2024b): for each of 13,452 headwater catchments (

EGUsphere Preprint repository

Wetterdienst, DWD) and represents a reprocessed version (Lengfeld et al., 2019) of DWD's operational radar-based quantitative precipitation estimation product, RADOLAN (Winterrath et al., 2012). The dataset has a spatial resolution of  $1 \times 1 \text{ km}^2$ , an hourly temporal resolution, and is openly available via the DWD open data server (Winterrath et al., 2018).

# 2.2 **DEM**

For catchment delineation and runoff analysis, we used the EU-DEM (European Commission, 2016), which has a 25 m resolution and combines SRTM (Shuttle Radar Topography Mission) and ASTER GDEM (Advanced Spaceborne Thermal Emission and Reflection Radiometer Global Digital Elevation Model).

# 2.3 Land use and soil data

Information on land cover was obtained from CORINE CLC5-2018 (BKG, 2018), which classifies high-resolution satellite imagery into 37 land cover classes for Germany following the European Environmental Agency (EEA) nomenclature. The classification considers objects with a minimum size of 5 ha and is updated every three years. Soil data were derived from the BUEK 200 national soil survey (scale 1:200,000; BGR, 2018), compiled from federal state surveys by the Federal Institute for Geosciences and Natural Resources (BGR) in cooperation with the National Geological Services (Staatliche Geologische Dienste). For each mapping unit, BUEK 200 provides areal fractions of dominant soil types along with detailed profile information, including texture, bulk density, and other key properties.

# 3 Methods

Much of the data and methodology for this study are detailed in Voit and Heistermann (2024b). Here, we briefly describe the hydrological model, further explain the flood frequency analysis, and the selection of local counterfactuals.

# 3.1 Modelling surface runoff

The hydrological model (Voit, 2024) was specifically tailored to simulate flash flood events in small- to medium-sized basins. A detailed model description is provided in Voit and Heistermann (2024b). During flash floods, surface runoff dominates (Marchi et al., 2010; Grimaldi et al., 2010), while evaporation and groundwater dynamics are negligible. Accordingly, the model comprises two modules. First, effective rainfall is estimated using the SCS-CN method (U.S. Department of Agriculture-Soil Conservation Service, 1972), which is widely applied in flash flood modeling (Gaume et al., 2004; Borga et al., 2007; Emmanuel et al., 2017). Second, the geomorphological instantaneous unit hydrograph (GIUH), derived from the DEM, represents the concentration of quick runoff from effective rainfall. The model's lightweight design allows the computation of large numbers of counterfactual scenarios. As it does not account for channel hydraulics or engineered structures, the analysis is restricted to headwater catchments smaller than 750 km². In our analysis, this corresponds to 13,452 sub-catchments with an mean area of 15.7 km².

**Figure 1.** Development of local counterfactuals: a) Catchment of Interest (CoI, green) and its 10 neighbor catchments (NCs, dark blue) in a 30 km neighborhood (light blue). b) Selecting the HPE which caused the highest annual runoff peak (red dot, c) in the NC (red box). d) Transposing the HPE from the NC to the CoI and modelling the resulting runoff (e)). This procedure is repeated for each NC and steps c-e) are repeated for each year.

#### 3.2 Design of local counterfactuals

Local counterfactuals are HPEs drawn from a neighborhood (TD) of a given catchment of interest (CoI) — the catchment to which the counterfactual scenarios are applied, and transposed to the CoI. In this study, all aforementioned 13,452 headwater catchments smaller than 750 km<sup>2</sup> are individually treated as a CoI, meaning that the following procedure is applied to each of these catchments (see also Fig. 1 for illustration):

- 1. For each CoI, we identified the ten most similar catchments located entirely within a 30 km buffer around the CoI. Similarity was quantified using a KDTree based on the following scaled catchment attributes: GIUH time to peak, GIUH standard deviation, GIUH unit peak discharge, mean slope, mean elevation, elevation standard deviation, area, and mean curve number (see section 3.1). We refer to these as *neighbor catchments* (NCs; see Fig. 1a).
- 2. For each of these NCs, we model the quick runoff from 2001 until 2023 (Fig. 1b). We then identify the annual maximum peak discharge for each of the 23 years (Fig. 1c).
- 3. From RADKLIM, we extract the data for the 23 HPEs which caused the annual maximum peaks in the NC (Fig. 1b) and transpose them from their original spatial position from the centroid of the NC to the centroid of the CoI, thereby creating spatial counterfactuals (Fig. 1d). We ensure that the CoI and all its upstream catchments will be completely covered by the HPE, by adding a 70 km buffer on each side of the RADKLIM subset (for better visualization we do not show the buffer in Fig. 1). To ensure a consistent soil moisture state we add a 14-day temporal buffer before the actual event. If the CoI has upstream basins, we additionally transpose the HPEs to the centroid of every upstream basin.
- 4. We model the surface runoff that these counterfactual HPEs would have caused in the CoI (Fig. 1e) and record the maximum counterfactual peak discharge values for each year. We repeat steps 3 and 4 for all NCs.

We hypothesize that the representativeness of counterfactuals for the meteorological processes governing the CoI generally decreases with the distance between the corresponding NC and the CoI. To test hypothesis, we compared four transposition domains (TDs): a 10 km buffer, a 30 km buffer, and two ring-shaped TDs with inner–outer radii of 30–60 km and 60–90 km around the CoI, respectively.

#### 140 3.3 GEV distribution

Under certain conditions, block maxima are GEV-distributed (Fisher and Tippett, 1928; Gnedenko, 1943). These conditions are met for precipitation (Coles, 2001) and discharge. The cumulative distribution function (CDF) of the GEV is defined

$$G(x) = \begin{cases} \exp\left\{-\left[1 + \xi\left(\frac{x-\mu}{\sigma}\right)\right]^{-1/\xi}\right\} &, \xi \neq 0\\ \exp\left\{-\left(\exp^{(z-\mu)/\sigma}\right)\right\} &, \xi = 0, \end{cases}$$
(1)

with location  $\mu$ , scale  $\sigma$  and shape  $\xi$ .

From the GEV distribution, return levels can be obtained for return periods that are even longer than the length of record. However, this extrapolation is uncertain with limited sample size (Coles, 2001). Our suggestion is, hence, to increase the sample size with local counterfactuals. To fulfill the requirements of the Fisher-Tippet-Theorem, all block maxima have to be drawn from the same statistical distribution. We choose a very small area as TD and then select HPEs within this TD based on the streamflow response of neighboring catchments that are very similar regarding slope, elevation, land use and the unit 150 hydrograph (see section 3.2). Based on this similarity of catchments and the small TD, we regard this first requirement as fulfilled. To fulfill the requirements of the Fisher-Tippet-Theorem, it is also paramount not to arbitrarily discard subsets of the data. More specifically, this mandates to include **all** annual maximum peak discharge values from an NC (instead of, e.g., just the highest one) to keep a consistent block size.

In FFA, special attention is given to the shape parameter  $\xi$ : a large shape parameter indicates a heavy tailed distribution where extreme events with high magnitude can occur. Especially when fitted to limited data points, the GEV distribution can produce implausible parameter estimates or "poor fits". For this reason we disregard catchments where one of the previous GEV distributions has a shape  $0 \ge \xi 

185

$$\rho_p(u) = \begin{cases} pu & , u > 0\\ (p-1)u & , u \le 0 \end{cases}$$

$$(2)$$

165 (3)

With  $u = z_n - q$ . For high non-exceedance probabilities p (which corresponds to a return period  $T = \frac{1}{1-p}$ ), it leads to a strong penalty for data points that are still higher than the modeled quantile  $(z_n > q)$ .

The quantile score is then computed for each non-exceedance probability *p*:

$$QS(y,q;p) = \sum_{i}^{n} \rho_p(y_i - q) \tag{4}$$

with p-quantile q (a return level corresponding to T), tilted check-function  $\rho_p(\cdot)$  and block maximum  $y_i$  obtained from the n factual peaks in the CoI.

We estimate the parameters of three GEV distributions that are fitted on different subsets of data and refer to them as follows:

- **GEV**<sub>Col</sub>: fitted only to the factual peaks from the Col.
- GEV<sub>NCs</sub>: fitted only to the counterfactual peaks from the NCs.
- GEV<sub>all</sub>: fitted to both factual peaks from the CoI and the counterfactual peaks from the NCs.

Essentially, GEV\_NCs and GEV<sub>all</sub> are the GEV variants that we introduce as competitors against the conventional GEV<sub>CoI</sub> which is exclusively based on information obtained in the CoI. In order to verify the added value of the new GEV variants, GEV<sub>CoI</sub> serves as a reference. For this purpose, we use the quantile skill score (QSS) which compares the QS of GEV\_NCs and GEV<sub>all</sub> (denoted QS\_NCs and QS<sub>all</sub>, respectively) to the QS of our reference GEV<sub>CoI</sub> (QS<sub>CoI</sub>) as follows:

180 
$$QSS_i = 1 - \frac{QS_i}{QS_{CoI}}$$
 with  $i \in \{NCs, all\}$  (5)

The QSS can take values between minus infinity and 1. Positive values indicate that the competing GEV (GEV\_NCs or GEV<sub>all</sub>) is superior to the reference.

As the quantile score (Eq. 4) is always computed for a specific return period T (or non-exceedance probability p), the QSS itself is obtained for specific values of T, too (20, 50, 100 and 200 years in this study), similar to Fauer and Rust (2023, Fig. 4).

The reference  $QS_{CoI}$  itself is obtained by means of a leave-one-out cross-validation: to that end, one year i is excluded from the CoI's series of factual annual maxima and the  $GEV_{CoI}$  is estimated from the remaining training years. From the fitted  $GEV_{CoI,i}$ , a return level (p-quantile) is calculated and a quantile score  $QS_{CoI,i}$  is determined from this return level and the annual maximum value for year i. This is repeated for all years in the CoI series.  $QS_{CoI}$  is then obtained as the average of all  $QS_{CoI,i}$ .

190

# 4 Results and Discussion

# 4.1 Verifying the added value of counterfactual peaks on GEV estimation

**Figure 2.** Cumulative distributions showing the quantile skill scores for  $GEV_{NCs}$  in reference to  $GEV_{CoI}$ , for all subbasins and for four different transposition domains (10-km buffer: yellow, 30-km buffer:blue, 30-60-km ring: green, 60-90-km ring: orange. Subplots a)-d) show different quantiles that relate to the a) 20-year, b) 50-year, c) 100-year and d) 200-year flood. A quantile score > 0 indicates the superiority of the  $GEV_{NCs}$ . The median QSS of the 30-km buffer is indicated with the vertical blue dashed line

200

220

For each small-scale basin in Germany, we created local counterfactuals (section 3.2) and used these counterfactual flood peaks to fit a GEV distribution for the CoI. To validate how well local counterfactuals are able to represent the quantiles in the data (the factual flood peaks in the CoI), we performed an out-of-sample-test by comparing the  $GEV_{NCs}$  with the  $GEV_{CoI}$ .

The inspection of the GEV parameters for each catchment reveals a large number of implausible shape parameters, especially for GEV<sub>CoI</sub>, which is fitted to only 23 year-maxima. Because the number of data points increases by adding counterfactual peaks, the fits for GEV<sub>NCs</sub> improve: about 29 % of the basins cannot be included in the analysis because the implausible fit of GEV<sub>CoI</sub> whereas 5 % of the catchments have to be excluded because of an implausible fit of GEV<sub>NCs</sub> (see section 3.3). In total this leads to an exclusion of  $\approx$  33 % of the basins.

Figure 2 shows the results for all TDs and for four different return periods (20, 50, 100 and 200 years). Since negative values of the QSS are harder to interpret, we show only QSS >= 0. We will discuss the differences between the different TDs in the following section. For now, we focus on the TD with a radius of 30 km. The main result is that the GEV<sub>NCs</sub> - which has never seen any information from the CoI - clearly outperforms the GEV<sub>CoI</sub>: across all return periods and transposition domains, the majority of catchments have positive QSS values. E.g., for the TD with a 30 km buffer, the percentage of catchments with positive QSS<sub>NCs</sub> values (see intercept on the y-axis) is 87% for T=20 a, 78% for T=50 a, 73% for T=100 a, and 69% for T=200 a. Evidently, GEV<sub>NCs</sub> performs worse for the corresponding remainder to 100%.

We would like to take a closer look at the differences between the return periods. Increasing return periods lead to a decreasing fraction of catchments with positive QSS<sub>NCs</sub> values - obviously not desirable -, but also to a desirable increase of catchments with very high QSS values (for  $T=20 \, a$ , 0.2% of the catchments have a QSS > 0.5, while this fraction grows to 28% for  $T=200 \, a$ ). Altogether, the median QSS continuously grows from a value of 0.16 for  $T=20 \, a$  to a value of 0.27 for  $T=200 \, a$ , suggesting that the value added by using GEV<sub>NCs</sub> increases with the return period. This is plausible, since return levels for low return periods can be estimated more robustly from short time series (for  $T=20 \, a$ , the estimation of a return level from an annual series of 23 years doe not even imply extrapolation).

These results serve as a proof of concept: for the majority of cases, we are able to better represent the quantiles in the data of the CoI by using a GEV distribution fitted exclusively to the counterfactual peaks ( $GEV_{NCs}$ ). The improvement is more pronounced for higher quantiles (or return periods). In practice the GEV would be fitted to both factual *and* counterfactual peaks together ( $GEV_{all}$ ), which only marginally increases the robustness of the return level estimates. The QSS for  $GEV_{all}$  is shown in Figure S1 in the supplement.

#### 4.2 Effect of different transposition domains

We calculated the  $QSS_{NCs}$  for four different TDs. This way, we want to investigate whether HPEs transposed from larger distances are less "typical" for the CoI and will therefore result in less representative GEV fits with lower values of  $QSS_{NCs}$ . This effect can be observed in Figure 2. For each return period, the intercepts of the QSS distributions on the y-axis are higher for the ring-shaped TDs (30–60 and 60-90 km) than the intercepts of the TDs with a 10- or 30 km buffer. This effect is less pronounced with increasing quantiles. The differences between the 10- and 30 km-buffers are very small. These results support the hypothesis that HPEs transposed over short distances are more representative for the HPEs occurring directly over the

CoI. Nevertheless, the sampling process of the NCs can also have an impact on the results. Within the TD we sample ten catchments which are most similar to the CoI (section 3.2). If the TD is very small, there are less basins to sample from so that the representativeness of the sampled HPEs for the CoI might suffer. Likewise it could also be possible that basins are less similar to each other with increasing distance. Due to the complex topography around every catchment, we think that there can be hardly a generalized solution for the "perfect" transposition domain. However, our results show, that there is, for most small-scale basins in Germany, no large difference whether the TD is a 10-, or 30 km buffer. Providing large computational resources this could be systematically investigated further by increasing the size of the TD step by step and evaluating the QSS.

# 4.3 Return levels

We would now like to demonstrate how the use of local counterfactuals affects return levels, in comparison to the conventional use of factual discharge peaks in the CoI. While GEV<sub>NCs</sub> was used for verification in section 4.1, we will now use GEV<sub>all</sub> there is not reason to entirely discard the data from the CoI data for GEV fitting. For the 200-year return period, Figure 3 shows the ratio between the return level obtained from  $GEV_{all}$  and from  $GEV_{Col}$  (as a histogram over all analysed catchments). For all TDs, the median ratio is very close to one, so using local counterfactuals results in lower return levels for half of the basins and to higher return level for the other half. In our view, this is an important insight: in contrast to our intuitive expectation, the use of local counterfactuals for GEV fitting does not systematically increase the resulting return levels, but simply reduces the estimation error over all CoIs (based on the higher QSS and the narrower confidence intervals, see below). However, this improvement of the GEV estimation is still based on the inclusion of higher discharge maxima via counterfactuals. This is illustrated by the gray histograms in Fig. 3 which show, for each catchment, the ratio between the highest value in the annual maximum series of counterfactual and factual peaks and the highest value in the annual maximum series of just the factual peaks. The gray histograms clearly show that counterfactuals increase the maximum of the complete series of annual maxima, leading to more robust GEV fits. The medians for all TDs are between 1.43 and 1.6. It is important to note that the four TDs cover very different spatial extents: the 30-60-km-ring has an area of 14,137 km<sup>2</sup>, while the 10 km buffer has a size of ~466 km<sup>2</sup> (for a circular basin with an area of 15 km<sup>2</sup>). The larger the TD we sample HPEs from, the more options we have to find catchments which are very similar to the CoI. Thus, it is also more likely that we are sampling HPEs that matter regarding the formation of an extreme flood peak in the CoI. This absolute maximum peak could serve as reference for the probable maximum flood (PMF) and is automatically included in the results of the analysis. Yet, remember that sampling from such more distant and larger neighborhoods does not improve the GEV estimation, as was shown in section 4.2.

The difference between the two medians shown in Figure 3 may appear counterintuitive. However, two factors account for this observation. First, although the counterfactual dataset exhibits some higher peaks, these peaks occur jointly with the entire set of annual maxima from this NC (23 values). In many cases these high peaks have little impact on the GEV fit due to the amount of data points that are just "average" peaks. Figure 4 shows an example of this case. The sampling error with small sample sizes as the 23 annual maxima for the GEV<sub>CoI</sub> can lead to very heavy tailed GEV fits, high return level estimates and very wide confidence intervals. Even though the data pool for GEV<sub>all</sub> (253 values) contains more extreme peaks (max. CoI:

Figure 3. Histogram of the ratio between the 200-year return level from  $GEV_{all}$  and from  $GEV_{Col}$  for four TDs. Gray histograms indicate the ratio of the highest peaks in the respective datasets used for fitting (see main text for further explanation). The median ratio of the return levels ratio is marked in black, and the median ratio of maximum peak discharge values in red.

43.2  $m^3s^{-1}$ , max. all: 87.3  $m^3s^{-1}$ ), the fit is still mainly influenced by the larger amount of moderate peaks and results in lower return level estimates.

Secondly, local counterfactuals also induce spatial smoothing (which is desired): each catchment is a CoI once, but serves as neighbor for many other CoIs. As a result, nearby and hydrologically similar catchments often share almost identical sets of peaks. When a counterfactual peak increases the return level estimate for one CoI, the peaks from that CoI will also enter the

**Figure 4.** Comparison of two  $GEV_{Col}$  and  $GEV_{all}$  for one exemplary basin. a) Return levels estimated by  $GEV_{all}$  (orange) are lower than by  $GEV_{Col}$  (purple). The shaded areas mark the 95 % confidence interval estimated with boot strapping (n=500). The empirical return periods were estimated with the Weibull plotting position and are indicated with the semi-transparent dots. b) Density histogram of the annual maxima and fitted GEV distribution.

NC data pool once their roles are reversed. In this case, the inclusion of the peak can reduce the return level estimate for the neighboring catchment.

The estimation of return levels beyond the observational period comes with large uncertainties in the case of  $GEV_{CoI}$  in the example in Fig. 4: the 200-year return level in is between 34 and 325  $m^3s^{-1}$  (95 % confidence interval). This range is much smaller for  $GEV_{all}$ , where the 200-year return level is between 87 and 130  $m^3s^{-1}$ . Across all catchments, the 95 % confidence

intervals shrink substantially. Within the 30 km buffer, the median reduction in interval span is 78.75 % for the 20-year return level, 86.25 % for the 50-year level, 89.75 % for the 100-year level, and 92.25 % for the 200-year level.

#### 270 5 Limitations

The main limitation in this study is the short observational period. The radar-based precipitation dataset RADKLIM covers 23 years. For the computation of the QSS, the 23 annual maximum flood peaks, which were modelled, using RADKLIM as forcing, served as the verification for  $GEV_{CoI}$  and  $GEV_{NCs}$ . Even though it would be desirable to have more data for this comparison, there is simply no stream or rain gauge data available for the majority of small-scale basins in Germany. Stream gauge observations are rarely longer than 30 years, so our experiments were designed upon the practical availability of data. Our results underline that using the GEV based on little observations has to be done with great care. We had to dispose 31 % of GEV fits, mainly because of the small sample size for the  $GEV_{CoI}$ .

The HPEs transposed to create local counterfactuals are selected based on the discharge peaks they generate in the NC. This approach avoids the need to predefine the characteristics of a relevant HPE prior to selection. However, discharge-relevant HPEs may be overlooked—for instance, in cases where an extreme HPE precipitates across a watershed. To at least partially address this limitation, we additionally transposed HPEs to every subbasin within larger CoIs, thereby capturing a greater degree of spatial variability.

Finally, our hydrological model surely introduces uncertainty. We assume that catchment-specific biases of the simulated flood peaks to a certain degree cancel out when comparing return periods and return levels. More importantly, our hydrological model was chosen to systematically conduct our study for all of Germany and for a large amount of catchments. The presented *concept* of using local counterfactuals for GEV estimation is independent of the hydrological model. For practical applications, e.g. in agencies in charge of risk management or design of hydraulic infrastructure, we would recommend to repeat the analysis with a hydrological model that is calibrated and validated to the local conditions.

## 6 Conclusions

In this study, we introduced a framework to increase the robustness of the GEV fits for flood frequency analysis by utilizing local counterfactuals. While being inspired by the concept of stochastic storm transposition, we follow a different approach in selecting candidate HPEs (based on the discharge response they caused in hydrologically similar neighbor catchments within a specific search radius around the the CoI), and in transposing these candidate events within the transposition domain (not stochastically, but systematically right over the CoI).

In a case study for Germany, we provided a proof-of-concept by applying this framework to a set of  $\approx 13,452$  catchments smaller than  $750 \, \mathrm{km}^2$ . For that purpose, we combined 23 years of radar-based precipitation records with a Germany-wide flash flood model. By using the quantile skill score, we verified that the use of local counterfactuals improves the fit of GEV

325

parameters for the vast majority of catchments. As expected, the value added by this approach increases with the return period of interest.

The main advantage of this approach the increased precision of the GEV return level estimates with much narrower confidence intervals. This is especially relevant for floods with return periods beyond the observational period. According to the Floods Directive of the European Union (2007/60/EC, (European Commission, Directorate-General for Environment, 2013)), this is particularly relevant for floods of "medium probability" (T=100 a) and floods of low probability (which in Germany is defined as a flood with T=200 a). We could show that, across return periods, the use of local counterfactuals improves GEV fitting, but does not lead to a systematic change of return levels across the entirety of investigated catchments. Furthermore, our approach also automatically yields the worst-case flood.

The selection of the TD affects the quality GEV estimation when local counterfactuals are employed. We showed that the QSS decreased when HPEs were sampled from a distance of more than 30 km away from the CoI. Still, the optimal definition of the TD will remain arbitrary and represents a subject for further research, as it represents an inherent trade-off: while an increasing distance allows us to sample from a larger variety of events and particularly from a larger choice of hydrologically similar catchments, an increasing distance will typically sample HPEs that are less representative for the meteorological processes that govern the CoI. At of now, the 30 km radius remains a rather pragmatic choice and a compromise between these two requirements.

The practical application of our framework appears suitable for all contexts in which observational records are short in comparison to the return period required for a specific purpose, such as land use planning, design, or insurance. For such applications, we strongly recommend to use a hydrological model that is calibrated and validated for the local or regional conditions.

#### Code and data availability.

We published notebooks and code which demonstrate our hydrological model for a small, exemplary region (Altenahr basin): the derivation of GIUHs from a digital elevation model, the extraction of rainfall data from and effective rainfall for the subbasins from RADKLIM data and the modelling of quick runoff. The code is published at: https://doi.org/10.5281/zenodo. 10473424.

All data used in this study is accessible at the open data repository of the DWD: the RADKLIM\_RW\_2017.002 dataset is available at https://opendata.dwd.de/climate\_environment/CDC/grids\_germany/hourly/radolan/reproc/2017\_002, (Winterrath et al., 2018); the EU-DEM is available at https://ec.europa.eu/eurostat/web/gisco/geodata/digital-elevation-model/eu-dem# DD, (European Commission, 2016); the CLC5-2018 land cover data is available at https://gdz.bkg.bund.de/index.php/default/open-data/corine-land-cover-5-ha-stand-2018-clc5-2018.html, (BKG, 2018). The soil data is available at https://www.bgr. bund.de/DE/Themen/Boden/Informationsgrundlagen/Bodenkundliche\_Karten\_Datenbanken/BUEK200/buek200\_node.html, (BGR, 2018) All data last accessed 27 June 2024.

330 *Author contributions*. PV, FF, and MH conceptualized this study. PV carried out the analysis, produced the figures and wrote the manuscript, with contributions from FF and MH.

Competing interests. The contact author has declared that neither they nor their co-authors have any competing interests.

Acknowledgements. Paul Voit and Felix Fauer were funded by the ClimXtreme program of the German Federal Ministry of Research, Technology and Space (PV: grant number 01LP2324B; FF: Grant number 01LP2323H).

We would like to thank the open-source community; without its software and data this study would have not been possible. Some small parts of the text were improved in exchange with a language model (https://chat.openai.com/chat, last access: 7 October 2025).

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
