# Peer review of "From Worst-Case Scenarios to Extreme Value Statistics: Local Counterfactuals in Flood Frequency Analysis"

_EGUsphere, 2025_

## Author Comment (AC1)

**Interactive Discussion: Author Response to Referee #1**

**From Worst-Case Scenarios to Extreme Value Statistics: Local Counterfactuals in Flood Frequency Analysis**

Paul Voit, Felix Fauer, and Maik Heistermann
*NHESS Discussions,* `doi:10.5194/egusphere-2025-4951`

RC: *Reviewer Comment*,    AR: *Author Response*,    ☐ Manuscript text

Dear Referee,

thank you for taking the time and effort to review this manuscript. Surely, incorporating your suggestions will improve the readability and overall quality of the manuscript.

Please find our responses to your comments below. These should be considered as preliminary (part of the interactive discussion) since the actual implementation of changes depends on the editorial decision.

Thanks again for your efforts!

Kind regards,
Paul Voit, Felix Fauer, and Maik Heistermann

**RC:** *The term "local counterfactual" should be defined more firmly in the opening section so that readers unfamiliar with the concept can immediately grasp its hydrological meaning.*

AR: We agree and try to address the comment by the following changes:

In the sentence after l. 62 of the preprint:

> [...] Voit and Heistermann (2024a) introduced the concept of "local counterfactuals": they selected HPEs that had caused high runoff peaks in basins from a close (i.e. "local") neighborhood around the CoI (more specifically, a 20 km radius), transposed these events to the CoI and used it to force a rainfall-runoff model that would than return the counterfactual flood peak. The approach was based on the assumption that if an HPE were sampled from a local neighborhood, it would be more representative for HPEs that are "typical" for the CoI. Even with this local TD, local counterfactuals produced flood peaks comparable to a 200-year return level flood.

**RC:** *The introduction would benefit from a clearer explanation of why a 30-km neighborhood and ten neighboring catchments were selected as the basis for local counterfactual generation.*

AR: While the size of the transposition domain (30 km neighborhood) will most likely always remain a matter of discussion, it would be generally preferable to select more than 10 neighboring catchments and thus, create

more counterfactuals. This decision was mainly based on computational limitations because every additional neighboring catchment results in additional 23 counterfactuals which need to be modelled. We will clarify this by adding the following sentence in line 85:

> A 30 km radius neighborhood (transposition domain) can be still be considered as local and small, compared to the domain sizes in other studies (e.g. Voit and Heistermann, 2024; Abbasian et al., 2025) while the number of 10 neighboring catchments was chosen mainly to contain the computational load.

**RC:** *The background section is strong, but it would be helpful to distinguish more explicitly between catchment similarity and storm similarity, as the manuscript presently assumes these are equivalent.*

AR: We entirely agree, and we will attempt, in the revised manuscript, to clarify the notion of similarity already in the introduction. Most importantly, we will try to better explain that the use of catchment similarity metrics is, to a considerable extent, *also* motivated by the aim to identify similar storms. How is that? Surely, the 30 km radius is the prime filter to make sure we sample storms from an atmospheric environment that is governed by similar mechanisms as the CoI. Yet, sampling storms that caused annual maxima in similar catchments ensures that the sampled storms have spatio-temporal characteristics that make them impact-relevant for the CoI (e.g. similar size or similar unit hydrographs) and that could also occur over the CoI given the potential for orographic effects (e.g. similar mean and standard deviation of elevation in the catchment). Based on these considerations, we aim to create counterfactuals that are representative for our CoI. In the revised manuscript, we will extend the corresponding explanations on the design of the local counterfactuals, and we will also discuss in further depth the limitations that we face in creating such representative counterfactuals.

**RC:** *The use of an uncalibrated SCS-CN and GIUH model across more than 13,000 catchments introduces considerable uncertainty, and the authors should include either a brief validation example or a reference to previous calibration results.*

AR: We agree that our hydrological model introduces considerable uncertainty, as would any hydrological model under extreme rainfall-runoff conditions. However, our modelling approach is well established in the flash flood community (Marchi et al., 2010; Borga et al., 2007; Ruiz-Villanueva et al., 2012; Tarolli et al., 2013). Furthermore, Voit and Heistermann (2024) could show for the Ahr flood in 2021 in Western Germany that our model was able to reproduce the reconstructed flood hydrograph at gauge Altenahr (Roggenkamp and Herget, 2022; Mohr et al., 2023) very well (Fig. 1). For that reason, we think that the presentation of additional validation results is not required within the present manuscript. This is also because we only compare model results within each catchment to one each other. Within such a comparison, any systematic model errors should tend to cancel out when used for GEV fitting. Furthermore, our study should be considered as a proof-of-concept. In the section "Limitations", we also recommended, for practical applications in the context of risk management, to rather use a model that has proven valid for the application region.

However, we suggest to expand (also in response to other referee comments) the section on 'Limitations" further with regard to the uncertainties of the hydrological model and potential implications for our analysis.

**RC:** *The criteria used to define "catchment similarity" deserve more explanation, especially regarding how the attributes were scaled and weighted in the KDTree analysis.*

AR: In our analysis, we aimed at sampling rainfall events that would have a strong impact in the CoI. Because we based the selection on the flood peaks in the NCs, we want to ensure that the catchments are hydrologically similar, as well as that the rainfall events are representative for the factual rainfall events in the CoI. For this reason, we based similarity mostly on descriptors of topography, land use and soil which should i) govern the

[Figure]

Figure 1: Top: Reconstruction by Roggenkamp and Herget (2022) published in Mohr et al. (2023) of the flood event in Altenahr, West-Germany, July 2021. Discharge is shown in red, water level in blue. Bottom: Modelled discharge for the same event with RADKLIM data with our model.

formation and concentration of surface runoff and ii) ensure that potential orographic effects could occur both in the CoI and the NCs. Following descriptors were chosen:

- Peak [m³/s], time to peak [s] and standard deviation [m³/s] of the unit hydrograph: The unit hydrograph is derived directly from the DEM, similar hydrographs imply, to a certain degree, similar topography.

- Upstream catchment area

- Curve number (soil moisture class 2): The curve number represents soils and land use in our model. A similar curve number would lead to a similar runoff generation in our model.

- Mean and standard elevation of the DEM and mean slope. With this descriptor we try to avoid sampling rainfall events from catchments which are e.g. situated at a substantially different elevation. If the CoI

was e.g. close to a mountain range, rainfall events should not be sampled from this mountainous area, because they might not be representative for the rainfall events occuring in the CoI.

- Unit Peak Discharge: The peak of the unit hydrograph divided by the catchment area is yet another descriptor of the hydrological character of the catchment.

We used the KDTree-algorithm from the Python library "SciKit-Learn" and scaled all catchment descriptors with the "StandardScaler" from this library to ensure that none of this descriptors dominates the decision for similarity.

We suggest to change following sentence in section 3.2, line 123, to clarify the scaling:

> Similarity was quantified using a KDTree (SciKit-Learn) based on the following scaled (Scikit-Learn StandardScaler) catchment attributes:...

**RC:** *The assumption that storms producing high runoff in a nearby basin are hydrologically meaningful for the catchment of interest should be justified with either empirical evidence or literature support.*

AR: We are not entirely sure what the referee means by "hydrologically meaningful for the catchment of interest". We assume, however, the referee means that the selected storms should be "representative" for the kind of storms that cause flood peaks in the CoI. In that regard, we would like to refer to our above explanations on similarity. At the same time, the referee implies that all our assumptions on similarity and hence representativeness are only that: assumptions, or, more benevolent, "expert guess". That is correct. And while we think that our assumptions are plausible and well in line with "hydrological common sense", the only way to actually assess the validity of our similarity metrics is to compare them against others in a kind of benchmark analysis: considering our KDTree-approach as a filter, the question would be which filter provides the best results in terms of improvement of our performance metric (QSS). That way, we could at least say which filter is superior over another one. In fact, we did this when we analysed the sensitivity of the QSS to different neighbourhood radii around our CoI. But while it would be highly interesting to expand such an analysis to other similarity metrics, we think that this is beyond the scope of the present study which rather aims to introduce a framework and provide proof-of-concept. We will, however, expand our discussion of the limitations of similarity metrics, and also outline future research to assess the validity of similarity metrics.

**RC:** *The manuscript should explain how independence among counterfactual annual maxima is ensured, given that neighboring catchments may experience correlated rainfall events.*

AR: Thank you for this remark. Indeed, we make the assumption that the counterfactual HPEs represent alternative variants of a given HPE in the CoI, that could have happened at another time within the CoI. With this approach we increase the sample size to improve the GEV parameter estimation. Also, we argue that events which cover two or more NCs at the same time, are allowed to have more influence on the GEV parameter estimation.

**RC:** *Mixing factual and counterfactual peaks in a single GEV fit may violate standard assumptions, and this issue requires at least a clear justification in the methods section.*

AR: Thank you for this comment. We agree that further justification might improve the manuscript. Since the peaks of factual and counterfactual HPEs are determined with the same method, we argue that both can be pooled to fit a GEV. The discriminating characteristic between factual and counterfactual is that counterfactual peaks are derived from storm transposition.

We suggest to add in line 153:

> Since the peaks of factual and counterfactual HPEs are determined with the same method, both can be pooled to fit a GEV, given all assumptions above.

**RC:** *Although the QSS results show improvements, the authors should comment on the fact that GEV$_{NCs}$ outperforms GEV$_{CoI}$ even without using any data from the catchment of interest, which may indicate over-smoothing or strong regional influences.*

[Figure]

Figure 2: Cumulative distributions showing the quantile skill scores for GEV$_{NCs}$ in reference to GEV$_{all}$, for all subbasins and for four different transposition domains (10-km buffer: yellow, 30-km buffer:blue, 30-60-km ring: green, 60-90-km ring: orange. Subplots a)-d) show different quantiles that relate to the a) 20-year, b) 50-year, c) 100-year and d) 200-year flood. A quantile score > 0 indicates the superiority of the GEV$_{NCs}$. The median QSS of the 30-km buffer is indicated with the vertical blue dashed line

**AR:** Thank you for this comment. We believe that these results underline a strong regional influence and are a justification of our method. Apparently the flood peaks which we generated by sampling rainfall events from hydrological similar and nearby basins leads to counterfactual annual flood peaks that fit very well into the distribution of "observed" flood peaks. The uncertainty of the GEV fit decreases significantly when using 230 instead of 23 values. We already comment on the regional smoothing in section 4.3:

> Secondly, local counterfactuals also induce spatial smoothing (which is desired): each catchment is a CoI once, but serves as neighbor for many other CoIs. As a result, nearby and hydrologically similar catchments often share almost identical sets of peaks. When a counterfactual peak increases the return level estimate for one CoI, the peaks from that CoI will also enter the NC data pool once their roles are reversed. In this case, the inclusion of the peak can reduce the return level estimate for the neighboring catchment.

We also suggest to add another sentence in section 4.1 to explain the more robust fit of the GEV:

> These results serve as a proof of concept: for the majority of cases, we are able to better represent the quantiles in the data of the CoI by using a GEV distribution fitted exclusively to the counterfactual peaks ($GEV_{NCs}$). *Besides the fact, that the counterfactual peaks represent the distribution of CoI peaks well, the $GEV_{NCs}$ is also more robust because it is fitted to 230 values, instead of the 23 values used for $GEV_{CoI}$.* The improvement is more pronounced for higher quantiles (or return periods). In practice the GEV would be fitted to both factual *and* counterfactual peaks together ($GEV_{all}$), which only marginally increases the robustness of the return level estimates. The QSS for $GEV_{all}$ is shown in Figure S1 in the supplement.

As referee #2 pointed out, the supplement was not online. We apologize for that attach the figure here (Fig. 2 and will make sure, that the Supplement is properly uploaded with the revised version of the manuscript.

**RC:** *The improvement of $GEV_{NCs}$ with increasing return period is convincingly shown, yet the manuscript should discuss why the lower tail benefits less from the counterfactual approach.*

AR: Thank you, we will try to make this clearer. The higher the return period, the more we need to extrapolate and the higher the uncertainty will be. When using only 23 years of data for extreme value statistics the uncertainty for the 200-yr return level is very high. With more data we do not extrapolate. In the case of the $GEV_{NCs}$ we already have 230 (counterfactual)-"annual" maxima. The uncertainty for the 200-yr return level is very low, as shown in the example in Figure 4. We will add two sentences in line 212 to clarify this further.

> We would like to take a closer look at the differences between the return periods. Increasing return periods lead to a decreasing fraction of catchments with positive $QSS_{NCs}$ values - obviously not desirable -, but also to a desirable increase of catchments with very high QSS values (for T=20 a, 0.2% of the catchments have a QSS > 0.5, while this fraction grows to 28% for T=200 a). Altogether, the median QSS continuously grows from a value of 0.16 for T=20 a to a value of 0.27 for T=200 a, suggesting that the value added by using $GEV_{NCs}$ increases with the return period. This is plausible, since return levels for low return periods can be estimated more robustly from short time series (for T=20 a, the estimation of a return level from an annual series of 23 years does not even imply extrapolation). *The uncertainty increases the more we extrapolate beyond the length of the annual series. Especially for high return periods the benefit of an increased data basis is visible in these results.*

**RC:** *The discussion should reflect that counterfactual extremes depend strongly on the selected time window and may not represent the full range of possible events.*

AR: We agree. The longer the record length, the higher the probability it will contain an event with an even larger magnitude (although we would be careful with the term "full range of possible events" - even with very long

time series, we will have difficulties in spanning that range). In that sense, an analysis that includes local counterfactuals shows exactly the same behavior as conventional flood frequency analysis within the CoI. We will expand the section "Limitations" accordingly.

RC: ***The authors appropriately highlight the short time series, but they omit discussion of potential non-stationarity in rainfall over the 2001–2023 period, which may influence GEV tail behavior.***

AR: We agree that non-stationarity of the extreme value distribution is not accounted for by our approach, and we will expand the discussion of "Limitation" in order to point this out. That being said, we would speculate (meaning that we cannot prove it) that the brevity of the time series underlying conventional GEV fitting is a more important source of uncertainty than the non-stationarity of the distribution.

RC: ***The conclusion section accurately summarizes the study, but it should offer clearer guidance on when the counterfactual method might be unsuitable—particularly in regions with strong orographic gradients or highly heterogeneous rainfall patterns.***

AR: Thank you for this suggestion. We will add following part to the "Conclusions":

> The selection of the TD affects the quality GEV estimation when local counterfactuals are employed. We showed that the QSS decreased when HPEs were sampled from a distance of more than 30 km away from the CoI. Still, the optimal definition of the TD will remain arbitrary and represents a subject for further research, as it represents an inherent trade-off: while an increasing distance allows us to sample from a larger variety of events and particularly from a larger choice of hydrologically similar catchments, an increasing distance will typically sample HPEs that are less representative for the meteorological processes that govern the CoI. At of now, the 30 km radius remains a rather pragmatic choice and a compromise between these two requirements. In regions with high orographic gradients or highly heterogeneous rainfall patterns the size of the TD might have to be reduced or optimized in benchmark experiments similar to the one carried out in this study.

**References**

Abbasian, M., Wright, D. B., Notaro, M., Vavrus, S., and Vimont, D. J.: Flood frequency sampling error: insights from regional analysis, stochastic storm transposition, and physics-based modeling, Journal of Hydrology, p. 133802, 2025.

Borga, M., Boscolo, P., Zanon, F., and Sangati, M.: Hydrometeorological analysis of the 29 August 2003 flash flood in the Eastern Italian Alps, Journal of hydrometeorology, 8, 1049–1067, 10.5194/egusphere-2025-495110.1175/JHM593.1, 2007.

Marchi, L., Borga, M., Preciso, E., and Gaume, E.: Characterisation of selected extreme flash floods in Europe and implications for flood risk management, Journal of Hydrology, 394, 118–133, 10.5194/egusphere-2025-495110.1016/j.jhydrol.2010.07.017, 2010.

Mohr, S., Ehret, U., Kunz, M., Ludwig, P., Caldas-Alvarez, A., Daniell, J. E., Ehmele, F., Feldmann, H., Franca, M. J., Gattke, C., Hundhausen, M., Knippertz, P., Küpfer, K., Mühr, B., Pinto, J. G., Quinting, J., Schäfer, A. M., Scheibel, M., Seidel, F., and Wisotzky, C.: A multi-disciplinary analysis of the exceptional flood event of July 2021 in central Europe–Part 1: Event description and analysis, Natural Hazards and Earth System Sciences, 23, 525–551, 10.5194/egusphere-2025-495110.5194/nhess-23-525-2023, 2023.

Roggenkamp, T. and Herget, J.: Hochwasser der Ahr im Juli 2021–Abflussabschätzung und Einordnung, Hydrologie und Wasserbewirtschaftung, 66, 40–49, 2022.

Ruiz-Villanueva, V., Borga, M., Zoccatelli, D., Marchi, L., Gaume, E., and Ehret, U.: Extreme flood response to short-duration convective rainfall in South-West Germany, Hydrology and Earth System Sciences, 16, 1543–1559, 10.5194/egusphere-2025-495110.5194/hess-16-1543-2012, 2012.

Tarolli, M., Borga, M., Zoccatelli, D., Bernhofer, C., Jatho, N., and Janabi, F. a.: Rainfall space-time organization and orographic control on flash flood response: the Weisseritz event of August 13, 2002, Journal of Hydrologic Engineering, 18, 183–193, 10.5194/egusphere-2025-495110.1061/(ASCE)HE.1943-5584.0000569, 2013.

Voit, P. and Heistermann, M.: A downward-counterfactual analysis of flash floods in Germany, Natural Hazards and Earth System Sciences Discussions, 2024, 1–23, 10.5194/egusphere-2025-495110.5194/nhess-2023-224, 2024.

---

## Author Comment (AC2)

**Interactive Discussion: Author Response to Referee #2**

**From Worst-Case Scenarios to Extreme Value Statistics: Local Counterfactuals in Flood Frequency Analysis**

Paul Voit, Felix Fauer, and Maik Heistermann

*NHESS Discussions,* `doi:10.5194/egusphere-2025-4951`
* * *
**RC:** *Reviewer Comment*,    AR: *Author Response*,    ☐ Manuscript text

Dear Referee,

thank you for taking the time to review this manuscript. Your suggestions and feedback will surely improve this manuscript.

Please find our responses to your comments below. These should be considered as preliminary (part of the interactive discussion) since the actual implementation of changes depends on the editorial decision.

Thanks again for your efforts!

Kind regards,
Paul Voit, Felix Fauer, and Maik Heistermann

Main comments:

**RC:** *From a comprehensive analysis of 13000 catchments, I was hoping to see a map with the regional performance of this method, and the areas that might be problematic.*

AR: We did not include a map because we could not distinguish any spatial patterns. We include here the map for the QSS for q=0.99 and q=0.995 (100-yr and 200-yr flood) for the 30km-buffer in Figure 1 and 2. Still, we think that these figures would not really add to the paper, so we would rather not include them in the revised version.

**RC:** *What is the effect of the catchment area on method performance and the size of the transposition domain?*

AR: This is a very interesting and thoughtful comment. We originally had expected that small catchments would benefit more from our counterfactual approach (in terms of QSS) since the likelihood to be "hit" by small convective rainfall events decreases with catchment size. Instead, we do see a larger improvement of the QSS for larger basins in Figure 4 (but note that there are only few basins larger than 40 km² in our model setup, see Fig. 3). This might be caused by the generally larger amount of counterfactual peaks for the larger basins, because they often consist of several subbasins (and we move the rainfall event to the centroid of every subbasin of the CoI). The larger amount of counterfactual peaks might simply result in a more robust GEV fit (see Table 1).

[Figure]

Figure 1: Quantile skill score for German subbasins for the 30-km buffer and the 0.99 quantile. Catchments with an upstream area > 750 km² are excluded (white).

**RC:** *The model you applied seems to have multiple limitations, and I am wondering what's the effect on your results. You choose to apply a CN model over lumped (small) basins. This implies that the response is driven by the cumulated value of precipitation and its distribution in time, and not by its intensity. Hortonian runoff is not simulated. The spatial variability of precipitation is lost - might be ok if your subbasins are very small.*

AR: The subbasins were chosen to be small (see Fig. 3 in the attempt to compensate for the lumped nature of the model. We hence think that the resulting uncertainty is rather low. Surely, the SCS-CN approach introduces considerable uncertainty (as would any hydrological model under extreme rainfall conditions). Namely, its inability to represent overland flow from infiltration excess might be a relevant source of uncertainty. As a consequence of the referee's comment (and other referee comments), we will expand the section "Limitations" to discuss more comprehensively the uncertainties to be expected from our hydrological model. At the same time, we would like to emphasize that we essentially present an analysis *framework* and recommend, for practical applications by e.g. agencies, to use a hydrological that has proven valid in the region of application.

**RC:** *You represent only "quick runoff", while many of those catchments have annual maxima in winter, when soils are wet and slow runoff has a very high contribution on peaks.*

AR: For extreme events, we assume slow flow components to be negligible at the scale of small catchments. How-

[Figure]

Figure 2: Quantile skill score for German subbasins for the 30-km buffer and the 0.995 quantile. Catchments with an upstream area > 750 km² are excluded (white).

ever, it is correct that high flows may occur during winter subject to saturated soils and low evapotranspiration or in spring subject to snow melt events, and that these high flows may constitute some of the annual maxima and hence affect return levels for small return periods. As already mentioned above, we will expand the discussion of model uncertainties in the section "Limitations" based on this and other referee comments.

RC: *L121: In SST one of the most critical points is the definition of a similar transposition domain. With your method, you seem to transfer this to catchment similarity. Can you give more information on how the catchment similarity criteria are mixed, and how much your approach is sensitive to this choice?*

AR: Surely, the 30 km radius is the prime filter to make sure we sample storms from an atmospheric environment that is governed by similar mechanisms as the CoI. In that sense, we maintain the concept of a transposition domain, in analogy to SST. Furthermore, by sampling storms that caused annual maxima in similar catchments, we ensure that the sampled storms have spatio-temporal characteristics that make them impact-relevant for the CoI (e.g. similar size or similar unit hydrographs) and that could also occur over the CoI given the potential for orographic effects (e.g. similar elevation in the catchment). Based on these considerations, we aim to create counterfactuals that are representative for our CoI. In the revised manuscript, we will explain in more detail the corresponding similarity criteria and how they are combined ("mixed") by means of a KDTree-analysis: We used the KDTree-algorithm from the Python library "SciKit-Learn" and scaled all catchment descriptors with the "StandardScaler" from this library to ensure that none of this descriptors

[Figure]

Figure 3: Catchment area distribution of the 13,452 headwater catchments.

Table 1: Number of basins for the "30km-buffer" transposition domain and mean number of counterfactual peaks for each size class.

| size class | nr. of basins | mean nr. of counterfactual peaks |
|---|---|---|
| 0-20 km² | 5077 | 238 |
| 21-40 km² | 1625 | 378 |
| 41-60 km² | 732 | 564 |
| >61 km² | 1888 | 1240 |

dominates the decision for similarity. However, we acknowledge that many descriptors are correlated.

We will address to your second question ("how much [is] your approach sensitive to this choice?") in the following comment.

**RC:** *How sensitive is your method to not finding similar catchments in the transposition domain?*

AR: We would like to include in this answer the question from the previous comment ("how much [is] your approach sensitive to [the choice and combination of similarity metric]?"). Admittedly, the choice and combination of similarity metrics is a pragmatic one – an expert guess, if you will. But while we think that our assumptions are plausible and well in line with "hydrological common sense", the only way to actually *assess* the validity of our similarity metrics is to compare them against others in a kind of benchmark experiment: considering our KDTree-approach as a filter, the question behind such an experiment would be which filter could provide the best results in terms of improvement of our performance metric (i.e. the QSS). In fact, we did this when we analysed the sensitivity of the QSS to different neighborhood radii around our CoI.

[Figure]

Figure 4: Quantile skill score for German subbasins for all four transposition domains and the 0.995 quantile.

But while it would be highly interesting to expand such an analysis to the similarity metrics, we think that this is beyond the scope of the present study in which we rather aim to introduce a *framework* and provide proof-of-concept. We will, however, expand our discussion of the limitations of similarity metrics, and also outline perspectives for future research to assess the validity of similarity metrics.

**RC:** *One of your conclusions is that "the improvement declines when the radius of the transposition domain is extended beyond 30 km. I think this is based on Figure 2 alone. In this figure, performance seems to generally decline with larger radius, but I don't see a performance decline after 30 km2. I'd argue that the performance seems mostly independent from the size of the transposition domain (while it might be more sensitive to the measure of catchment similarity).*

AR: For the low return periods we can observe clear differences between the curves in Fig. 2 up to a QSS of 0.2. Generally speaking, CDF curves that are more shifted to the right indicate a better QSS. We hence do not quite agree with the referee that there is no performance decline after 30 km. However, we agree that the difference is becoming very small with increasing return periods.

Based on our results, we cannot refute the referee's hypothesis that the measure of catchment similarity is more important than the radius of the transposition domain. As pointed out in our response to the referee's previous comment, this would require an additional comprehensive benchmark experiment in which to investigate the sensitivity of the QSS to competing definitions of similarity. As already pointed out above, this is beyond the scope of the present study which rather aims to introduce a framework and provide proof-of-concept. We will, however, expand our discussion of the limitations of similarity metrics, and also outline future research

to assess the validity of similarity metrics.

**RC:** *Transposing the HPE to the centroid of the catchment might result in biases (e.g. if the storm and the NC have an orientation, or if spatial distribution over the catchment is important - see Zhou, 2021).*

AR: We agree and we have read the publication of Zhou et al. (2021). We would argue, that the spatial distribution and storm direction is increasingly important for larger catchments, e.g. where various tributaries flow together. Nevertheless, the elongation or orientation of a catchment could be included in the similarity criteria. As pointed out above, a systematic study in which different definitions/implementations of similarity are systematically benchmarked would be certainly worthwhile. We suggest to expand the section "Limitations" accordingly:

> Furthermore, the direction of the HPE relative to the shape of the catchment can have a major influence on the flood peak formation (Zhou et al., 2021). These indicators could be included in the descriptors of catchment similarity to improve the analysis.

Other comments:

**RC:** *"Worst-Case scenarios" is in the title and the conclusions, but what do you define as a "worst case"? Is taking the strongest HPE within 10km the worst that can happen to a basin? Sometimes compound scenarios or very high ( 500y?) return periods are considered.*

AR: We thank the referee for the comment. In fact, we think that the term "worst-case" is not required in the context of our study. Based also on other comments, we changed the manuscript title to "Considering rainfall events from a neighborhood to improve local flood frequency analysis". We will remove the sentence in which the term occurred in the manuscript's conclusions (ll. 305 ff. of the preprint) as it does not essentially relate to the subject and results of our study.

**RC:** *As you say, in your analysis FFA is strongly limited by the relatively short data availability (23 years). You apply a GEV over annual maxima using maximum likelihood, but usually for small data samples, POT and maybe L-moments estimation might be more appropriate.*

AR: We agree. We have started using the ML method in our first publications but have also realized that L-Moments is considered to be the more stable method. We also agree in regard to POT but this method is also not widely used by practitioners due to its more complex application, e.g. for the definition of the threshold. We are currently involved in a study in which we compare extreme rainfall events in southern India which are evaluated by an index based either on annual maxima or POT. Based on the results of that study, we might move to the POT method in the future to increase the robustness of our assessment. With regard to this manuscript, we suggest to add a brief discussion to the section "Limitations" in which we should point out that the use of a POT approach together with the Generalized Pareto distribution (GPD) might be preferable in case of limited time series lengths, and certainly compatible with the framework presented in our manuscript.

**RC:** *You use QS to evaluate particularly high return periods (200 years) over a short data record (23 years). If the quantile q is higher than every observations, is the best QS simply the closest to your observation? Is it correct to say that it's a better estimate?*

AR: Yes, exactly. Thank you for this remark. Indeed, if all observations are lower than the quantile, then the QS might reward the model that calculates the lowest quantile. Therefore, the QS is more reliable for the lower return periods, which are shown in Figs. 2 and 4. However, since the number of CoI is very large, some of them might experience observations above the 200-yr return period. Still, we suggest to add after line 184:

> Note that for very high return periods the QS might become unreliable, since only few or no observations are higher than the evaluated quantile. Then, the QS might just reward the model that predicts the lowest quantile.

Furthermore, we suggest to mention the limitation of the QS in section 5, "Limitations":

> The QS has to be handled with care for very high return periods. Observations that exceed high quantiles are rare and therefore, evaluating the model performance for unseen quantiles is challenging. This limitation applies to all scores known to us.

**RC:** *L26: I'm not sure why you describe flash floods here. You didn't specifically analyze effect on flash floods, and your approach is general.*

AR: We are using the term "flash floods" because we specifically look at small and medium sized catchments which are specifically prone to flash floods. It is also these catchments for which the lack of long observational time series particularly evident. Furthermore, the overall methodological setup with a rather small transposition domain is geared towards small to medium sized catchments.

**RC:** *L31: It's subjective, but I would not call a 750km2 basin "small". Maybe small + medium?*

AR: Yes, we agree.We suggest to change the sentence to: ...*flash-flood–prone basins generally small to medium sized (<1000 km²).*

**RC:** *L98: Isn't CORINE updated every 6 years?*

AR: Yes, at least according to the homepage. However, the most recent update is still the 2018 version (https://land.copernicus.eu/en/products/corine-land-cover).

**RC:** *L106: you refer to your other paper for the model application, but I think it would be useful to add some more information on the model setup and characteristics that are important for your results.*

AR: We suggest to change the section "Modelling surface runoff" as following:

The hydrological model (Voit, 2024) was specifically tailored to simulate flash flood events in small-to medium-sized basins. A detailed model description is provided in Voit and Heistermann (2024). During flash floods, surface runoff dominates (Marchi et al., 2010; Grimaldi et al., 2010), while evaporation and groundwater dynamics are negligible. Accordingly, the model comprises two modules. First, effective rainfall is estimated for each catchment and timestep (hourly) using the SCS-CN method (U.S. Department of Agriculture-Soil Conservation Service, 1972), which is widely applied in flash flood modeling (Gaume et al., 2004; Borga et al., 2007; Emmanuel et al., 2017). Since flash flood events predominantly occur during the summer months, we slightly adjusted the CN values for agricultural areas to account for the effects of summer crops (based on Seibert et al., 2020). A single CN value for each subbasin was then derived using an area-weighted average.

Second, the geomorphological instantaneous unit hydrograph (GIUH), derived from the DEM, represents the concentration of quick runoff from effective rainfall. The flow velocities were computed with the method of Maidment Maidment et al. (1996). This approach accounts for the increase in hydraulic radius with rising flow volumes, as described by Manning's equation, thereby capturing the downstream acceleration of flow without requiring the estimation of roughness coefficients for individual grid cells. In addition, it removes the need to distinguish between hillslope and channel grid cells within the catchment. The method assumes a velocity field that is invariant in both time and discharge, enabling the convolution of GIUHs to simulate the catchment response to the effective rainfall of an HPE. When two subcatchments converge, the hydrograph of the upstream basin is superimposed on that of the downstream basin with an appropriate time lag. This delay is defined by the travel time from the downstream basin's inlet to its outlet.

The model's lightweight design allows the computation of large numbers of counterfactual scenarios. As it does not account for channel hydraulics or engineered structures, the analysis is restricted to headwater catchments smaller than 750 km². Because of the lumped nature of the model it is crucial that the catchments are small enough to account for the spatial variability of rainfall. In our analysis, this corresponds to 13,452 sub-catchments with an mean area of 15.7 km² and a maximum headwater catchment size of 163 km².

**RC:** *L106: Please clarify: if I understand well, you apply a lumped CN model over basins with a median size of 15.7 km2. These basins are also combining into larger basins. I was confused how sometimes you talk about "upstream catchments" "transposition to each catchment in the CoI".*

AR: Yes, this is correct. We hope that this gets more clearer now with the suggested extended model description. If we look for similar NCs, we have to look at the total catchment area (including upstream basins) of both CoI and NCs. To clarify this, we suggest to change some parts in section 3.2:

For each CoI, we identified the ten most similar catchments located entirely within a 30 km buffer around the CoI. Similarity was quantified using a KDTree (SciKit-Learn) based on the following scaled (Scikit-Learn StandardScaler) catchment attributes: GIUH time to peak, GIUH standard deviation, GIUH unit peak discharge, mean slope, mean elevation, elevation standard deviation, *total catchment area (including the upstream subbasins)*, and mean curve number

and in line 132:

> If the CoI consists of various subbasin, we additionally transpose the HPEs to the centroid of every upstream subbasin.

**RC:** *L134: do you apply the HPE multiple times by transposing the same HPE to the centroid of each subcatchment? If so, do you think it's generates realistic precipitation fields?*

**AR:** Yes, this is what we are doing. We extract the HPE with a large spatial buffer so the whole HPE will always cover the CoI including all its upstream subbasins. The shifting distance between these subbasins is just a few kilometers due to the generally small size of the subbasins. We consider all these scenarios as realistic counterfactuals.

**RC:** *L157: do you mean that you disregard shape below 0 (0 is ok) and above 0.5?*

**AR:** Thank you for spotting this. It is of course the opposite to what is written in the manuscript. We checked our scripts and the correct version should be:

> For this reason only consider catchments where all of the previous GEV distributions have a shape parameter $0 \geq \xi < 0.5$

**RC:** *L195: GEV CoI is fitted over 22 years?*

**AR:** Yes, for the cross validation we only use 22 years. We repeat this 23 times and the take the average.

**RC:** *L217: I don't see the supplement, also online.*

**AR:** Indeed, it is not there. We apologize and are not sure about the cause for the missing supplement. We will make sure to upload it with the revised manuscript. For the time being, we put the information from the supplement here:

The figure in the supplement (Figure 5 here in the response letter) shows the results for all TDs and for four different return periods (20, 50, 100 and 200 years). According to Fauer et al. (2021) negative values of the QSS cannot be easily interpreted which is why we show only QSS >= 0. The inclusion of the data from the CoI improves the quantile estimation only marginally compared to $GEV_{NCs}$ (Fig. 2).

**RC:** *L220: are the HPE over larger domains less typical than (comment authors "for the"?) CoI or just less correlated? How much do the HPE of the CoI overlap with the floods over NC?*

**AR:** We chose two ring sized TPs to ensure that we actually only sample HPEs from further away. The hypothesis is, that these HPEs are less representative for the CoI, as pointed out in the manuscript. We did not check whether or not the HPE that caused the annual maximum over the CoI also caused the annual maximum over the NC.

**RC:** *L254: aren't NCs analyzed as a set? So with 230 values.*

**AR:** Sorry, this sentence is misleading. It hopefully becomes clearer if we write "23 for each NC". The total then is 230.

> First, although the counterfactual dataset exhibits some higher peaks, these peaks occur jointly with the entire set of annual maxima from this NC (23 values *for each NC*).

[Figure]

Figure 5: Cumulative distributions showing the quantile skill scores for $GEV_{NCs}$ in reference to $GEV_{all}$, for all subbasins and for four different transposition domains (10-km buffer: yellow, 30-km buffer:blue, 30-60-km ring: green, 60-90-km ring: orange. Subplots a)-d) show different quantiles that relate to the a) 20-year, b) 50-year, c) 100-year and d) 200-year flood. A quantile score > 0 indicates the superiority of the $GEV_{NCs}$. The median QSS of the 30-km buffer is indicated with the vertical blue dashed line

**RC:** *L258, L266, L267: I can't find those numbers reflected in figure 4. Am I reading it wrong?*

AR: Thanks again for your level of attention. We accidentally compiled an older figure into the PDF. This figure (Fig. 6 here in the response letter) is the correct one and will be implemented in the revised manuscript.

**RC:** *L284: why would catchment biases cancel out?*

AR: We will revise the manuscript in order to make the statement more comprehensible: if the model should have a systematic error (bias) in a specific catchment, than this bias should affect the peak discharge of all events simulated for that catchment and hence reflect in all the different GEV distributions fitted for that specific catchment. Apart from that, we will expand, as already stated in response to various referee comments, the section "Limitations" in order to more comprehensively discuss hydrological model uncertainties.

**References**

Borga, M., Boscolo, P., Zanon, F., and Sangati, M.: Hydrometeorological analysis of the 29 August 2003 flash flood in the Eastern Italian Alps, Journal of hydrometeorology, 8, 1049–1067, 10.5194/egusphere-2025-495110.1175/JHM593.1, 2007.

Emmanuel, I., Payrastre, O., Andrieu, H., and Zuber, F.: A method for assessing the influence of rainfall spatial variability on hydrograph modeling. First case study in the Cevennes Region, southern France, Journal of Hydrology, 555, 314–322, 10.5194/egusphere-2025-495110.1016/j.jhydrol.2017.10.011, 2017.

Fauer, F. S., Ulrich, J., Jurado, O. E., and Rust, H. W.: Flexible and consistent quantile estimation for intensity–duration–frequency curves, Hydrology and Earth System Sciences, 25, 6479–6494, 10.5194/egusphere-2025-4951/10.5194/hess-25-6479-2021, publisher: Copernicus GmbH, 2021.

Gaume, E., Livet, M., Desbordes, M., and Villeneuve, J.-P.: Hydrological analysis of the river Aude, France, flash flood on 12 and 13 November 1999, Journal of hydrology, 286, 135–154, 10.5194/egusphere-2025-495110.1016/j.jhydrol.2003.09.015, 2004.

Grimaldi, S., Petroselli, A., Alonso, G., and Nardi, F.: Flow time estimation with spatially variable hillslope velocity in ungauged basins, Advances in Water Resources, 33, 1216–1223, 10.5194/egusphere-2025-495110.1016/j.advwatres.2010.06.003, 2010.

Maidment, D., Olivera, F., Calver, A., Eatherall, A., and Fraczek, W.: Unit hydrograph derived from a spatially distributed velocity field, Hydrological processes, 10, 831–844, 10.5194/egusphere-2025-4951https://doi.org/10.1002/(SICI)1099-1085(199606)10:6<831::AID-HYP374>3.0.CO;2-N, 1996.

Marchi, L., Borga, M., Preciso, E., and Gaume, E.: Characterisation of selected extreme flash floods in Europe and implications for flood risk management, Journal of Hydrology, 394, 118–133, 10.5194/egusphere-2025-495110.1016/j.jhydrol.2010.07.017, 2010.

Seibert, S. P., Auerswald, K., Seibert, S. P., and Auerswald, K.: Abflussentstehung–wie aus Niederschlag Abfluss wird, Hochwasserminderung im ländlichen Raum: Ein Handbuch zur quantitativen Planung, pp. 61–93, 10.5194/egusphere-2025-495110.1007/978-3-662-61033-6$_4$, 2020.

U.S. Department of Agriculture-Soil Conservation Service: Estimation of Direct Runoff From Storm Rainfall, SCS National Engineering Handbook, Section 4, Hydrology. Chapter 10, 1972.

Voit, P.: A downward counterfactual analysis of flash floods in Germany – Code repository (v0.1), Zenodo [code], `https://doi.org/10.5281/zenodo.10473424`, last accessed: 15.08.2024, 2024.

Voit, P. and Heistermann, M.: A downward-counterfactual analysis of flash floods in Germany, Natural Hazards and Earth System Sciences Discussions, 2024, 1–23, 10.5194/egusphere-2025-495110.5194/nhess-2023-224, 2024.

Zhou, Z., Smith, J. A., Baeck, M. L., Wright, D. B., Smith, B. K., and Liu, S.: The impact of the spatiotemporal structure of rainfall on flood frequency over a small urban watershed: an approach coupling stochastic storm transposition and hydrologic modeling, Hydrology and Earth System Sciences, 25, 4701–4717, 10.5194/egusphere-2025-495110.5194/hess-25-4701-2021, 2021.

[Figure]

Figure 6: Comparison of two $GEV_{CoI}$ and $GEV_{all}$ for one exemplary basin. a) Return levels estimated by $GEV_{all}$ (orange) are lower than by $GEV_{CoI}$ (purple). The shaded areas mark the 95 % confidence interval estimated with boot strapping (n=500). The empirical return periods were estimated with the Weibull plotting position and are indicated with the semi-transparent dots. b) Density histogram of the annual maxima and fitted GEV distribution.

---

## Author Comment (AC3)

**Interactive Discussion: Author Response to Referee #3**

**From Worst-Case Scenarios to Extreme Value Statistics: Local Counterfactuals in Flood Frequency Analysis**

Paul Voit, Felix Fauer, and Maik Heistermann

*NHESS Discussions,* `doi:10.5194/egusphere-2025-4951`
* * *
RC: *Reviewer Comment*,     AR: *Author Response*,     ☐ Manuscript text

Dear Referee,

thank you for serving as referee for this manuscript and for your comments and suggestions. The quality of the manuscript will certainly improve on the basis of this review.

Please find our responses to your comments below. These should be considered as preliminary (part of the interactive discussion) since the actual implementation of changes depends on the editorial decision.

Thanks again for your efforts!

Kind regards,
Paul Voit, Felix Fauer, and Maik Heistermann

**RC:** *The paper would benefit from clearer definitions and a distinct conceptual separation between several key ideas that are currently addressed somewhat implicitly. In particular, the notion of "local counterfactuals" should be more rigorously defined early in the manuscript, including its hydrological interpretation and how it differs from related concepts such as storm transposition, spatial counterfactuals, and regionalisation. While the background section is strong, readers unfamiliar with these concepts may find it difficult to immediately understand what is novel versus what is adapted from existing approaches.*

AR: We suggest to add the following description of existing concepts to the introduction.

- **Regionalization**: Data from hydrologically similar catchments are incorporated into the estimation of distribution parameters to enhance the robustness of extreme value analysis (EVA) (e.g., Gaume et al., 2010; Guse et al., 2010; Nguyen et al., 2014; Halbert et al., 2016).

- **Probable maximum precipitation** (PMP): rainfall events from a "meteorological homogeneous" transposition domain are included in the analysis to increase the robustness (Fuller, 1914; District and Morgan, 1916) and to estimate the PMP (Hansen, 1987; WMO, 2009). Instead of exceedance probabilities this method only yields upper and lower bounds of precipitation. PMP can be used to estimate the upper bounds of a probable maximum flood (PMF), if used as forcing of a hydrological model. While PMP is widely applied in North America and Australia for designing high-risk infrastructure (e.g., dams and nuclear power plants), it is not prominently used in Europe. However, in recent years various studies regarding flood risk management have proposed and investigated different concepts of storm transposition, referring to the idea as "spatial counterfactuals" (Montanari et al., 2023; Merz et al., 2024; Voit and Heistermann, 2024; Vorogushyn et al., 2024; Thompson et al., 2025).

- **Stochastic storm transposition**: Building on the PMP/PMF concept, historical HPEs from a transposition domain are sampled using a Poisson distribution and randomly assigned (uniform distribution) within the domain, potentially affecting the catchment of interest (CoI). For flood frequency analysis, the resulting runoff in the CoI is simulated (e.g., Wright et al., 2014). This approach allows for the calculation of occurrence probabilities. For a detailed description see Wright et al. (2017). Globally, stochastic storm transposition (SST) remains rarely applied in practice (Wright et al., 2020) but it will form the core of the U.S. Federal Emergency Management Agency's "Future of Flood Risk Data" initiative, aimed at remapping the nation's floodplains (Abbasian et al., 2025).

- **Stochastic weather generators** are statistical models that simulate sequences of weather variables, such as temperature and precipitation, by randomly generating data based on observed patterns. They can be used to generate very long time series of meteorological forcings for a hydrological model (e.g. Falter et al., 2015; Apel et al., 2016).

**RC:** *The selection of a 30 km radius and ten neighboring catchments seems reasonable but is largely based on empirical judgment. The manuscript should offer a clearer rationale for these choices, whether based on meteorological homogeneity, hydrological similarity scales, or sensitivity analysis.*

AR: We agree that the choice and combination of similarity metrics is a pragmatic one – an expert guess, if you will. The only way to actually *assess* the validity of our similarity metrics is to compare them against others in a kind of benchmark experiment: considering our KDTree-approach as a filter, the question behind such an experiment would be which filter could provide the best results in terms of improvement of our performance metric (i.e. the QSS). In fact, we did this when we analysed the sensitivity of the QSS to different neighborhood radii around our CoI. But while it would be highly interesting to expand such an analysis to the similarity metrics, we think that this is beyond the scope of the present study in which we rather aim to introduce a *framework* and provide proof-of-concept. We will, however, expand our discussion of the limitations of similarity metrics, and also outline perspectives for future research to assess the validity of similarity metrics.

**RC:** *The criteria used to define catchment similarity via the KDTree deserve further explanation.*

AR: Yes, and your comment is in line with the other referees. The catchment similarity is a critical point. We

added more information (l. 123 ff) to further describe the process.

> We based similarity mostly on descriptors of topography, land use and soil which should i) strongly govern the formation and concentration of surface runoff and ii) ensure that potential orographic effects could occur both in the CoI and the NCs. Following descriptors were chosen:
>
> - Peak [m³/s], time to peak [s] and standard deviation [m³/s] of the unit hydrograph: The unit hydrograph is derived directly from the DEM, similar hydrographs imply, to a certain degree, similar topography.
> - Upstream catchment area
> - Curve number (soil moisture class 2): The curve number represents soils and land use in our model. A similar curve number would lead to a similar runoff generation in our model.
> - Mean and standard elevation of the DEM and mean slope. With this descriptor we try to avoid sampling rainfall events from catchments which are e.g. situated at a substantially different elevation. If the CoI was e.g. close to a mountain range, rainfall events should not be sampled from this mountainous area, because they might not be representative for the rainfall events occuring in the CoI.
> - Unit Peak Discharge: The peak of the unit hydrograph divided by the catchment area is yet another descriptor of the hydrological character of the catchment.
>
> We used the KDTree-algorithm from the Python library "SciKit-Learn" and scaled all catchment descriptors with the "StandardScaler" from this library to ensure that none of this descriptors dominates the decision for similarity. However, we acknowledge that some descriptors are correlated.

**RC:** *Applying an uncalibrated SCS-CN and GIUH-based lumped model across thousands of catchments introduces significant uncertainty. The model mainly captures fast runoff generation and overlooks Hortonian runoff, slow flow components, and spatial variability in precipitation. This is especially important for catchments with winter flood regimes, where soil saturation and slower processes may prevail. The authors should discuss how these simplifications could impact both annual maxima and GEV tail behaviour.*

AR: The referee is right. Uncertainties introduced by the hydrological model were already discussed in section 5 of the preprint, and we will expand this discussion based on the referee's comment. Here, we would like to maintain that for extreme events, we assume slow flow components to be negligible at the scale of small catchments. It is correct that high flows may occur during winter subject to saturated soils and low evapotranspiration or in spring subject to snow melt events, and that these high flows may constitute some of the annual maxima and hence govern return levels for small return periods. The tail behaviour of small catchments, however, is clearly dominated by convective heavy rainfall that occur in the summer. As for the effect of Hortonian surface runoff (infiltration excess), the referee is correct that this process is inadequately represented in the SCS-CN framework. We assume that this would lead to an underestimation of runoff generation for extreme events at very short durations, or, in other words, that a model that represents infiltration excess would lead to heavier tails of the GEV distribution.

**RC:** *The conclusions would benefit from clearer guidance on when the proposed method may be unsuitable.*

AR: Thank you, this was also mentioned by another referee. We added following sentence:

> In regions with high orographic gradients or highly heterogeneous rainfall patterns the proper size of the TD might have to be reduced or optimized in benchmark experiments similar to the one carried out in this study.

**RC:** *The manuscript frequently refers to "worst-case scenarios", but this term is not clearly defined.*

 AR: The manuscript mentions the term "worst-case flood" (or scenario) two times: in the title and in the conclusions. But the referee is completely right that it lacks clarity, an issue that was also noted by the other referees.

 In fact, we think that the term "worst-case" is not required in the context of our study. Based also on other comments, we changed the manuscript title to "Considering rainfall events from a neighborhood improves local flood frequency analysis". We will remove the sentence in which the term occurred in the manuscript (ll. 305 ff. of the preprint) as it does not essentially relate to the subject of our study.

**RC:** *The appropriateness of using annual maxima with maximum likelihood estimation for such short samples could be briefly discussed in relation to alternative approaches such as POT or L-moments.*

 AR: We agree. We have started using the ML method in our first publications but have also realized that L-Moments is considered to be the more stable method. We also agree in regard to POT but this method is also not widely used by practitioners due to its more complex application, e.g. for the definition of the threshold. We are currently involved in a study in which we compare extreme rainfall events in southern India which are evaluated by an index based either on annual maxima or POT. Based on the results of that study, we might move to the POT method in the future to increase the robustness of our assessment. With regard to this manuscript, we suggest to add a brief discussion to the section "Limitations" in which we should point out that the use of a POT approach together with the Generalized Pareto distribution (GPD) might be preferable in case of limited time series lengths, and certainly compatible with the framework presented in our manuscript.

**References**

[revised manuscript text omitted]